# Neural Network Ising Machines: Algorithm Unrolling for Combinatorial Optimization

## Abstract

We propose a new data-driven neural approach to combinatorial optimization in which we learn the parameters of an iterative dynamical system which efficiently samples good solutions for typical instances of the NP-hard Max-Cut/Ising problem. The dynamical system is parameterized by a small neural network which is trained using a zeroth-order optimization method. We find that our method is able to learn efficient and scalable algorithms for solving these combinatorial optimization problems. We show that even with a limited parameter count, the neural network is able to learn sophisticated dynamics which allow it to efficiently navigate the non-convex landscapes that are characteristic of NP-hard problems. We compare our method against state-of-the-art neural-CO approaches as well as other classical Max-Cut/Ising algorithms and show that is can achieve competitive performance.

## 1 Introduction

Combinatorial optimization (CO) problems are a class of problems which have important applications across many fields of science and engineering. However, because they are NP-hard there is no general algorithm which can solve these problems efficiently. Thus many heuristics and approximate algorithms have been developed which are effective on certain classes of CO problems. On the other hand, the techniques of neural networks and machine learning have been widely successful at learning patterns from data. This raises a central question: can heuristic algorithms for combinatorial optimization be learned from data, and can they ultimately outperform their handcrafted counterparts? This intersection between the two fields is attractive in a number of ways. First of all, many heuristic algorithms for CO are not well understood and are a result of a large amount of human experimentation and parameter tuning already. In this context it makes sense that an ML technique would be appropriate to further automate this process. Secondly, from the perspective of machine learning, one drawback that many neural network based algorithms have is that their output can be unreliable and unverifiable. This problem is partially avoided in the context of CO because the solutions we are looking for are inherently verifiable. For these reasons (among many others) the intersection of these fields, often referred to as "neural CO", has been extensively studied over the last decade or so (see section 2.1). However, although there have been many success stories of these methods, there is still no generally agreed upon technique, and in many cases hand-crafted classical heuristics are still better, making the field an area of active research. In this work, we propose a new method which optimizes the NP-hard Ising/Max-Cut problem and related problems. Our approach is closely related an recent line of work which aims at developing a dynamical systems approach to solving the Ising problem (see section 2.2), however we extend it in a data-driven matter. Although our method differs in both architecture and training method from existing neural CO approaches, we show that it can achieve state-of-the art performance on many commonly used benchmarks.

## 2 Related Works

### 2.1 Neural Combinatorial Optimization

In the field of neural combinatorial optimization (neural CO) many different types of CO problems have been considered, The most common of them being the traveling salesmen problem (TSP)

Vinyals et al. (2017); Bello et al. (2017); Deudon et al. (2018); Joshi et al. (2019); Bresson & Laurent (2021); Bogyrbayeva et al. (2022); Sui et al. (2024); Alanzi & Menai (2025). Many architectures have been used on TSP including pointer networks Vinyals et al. (2017), transformers Kool et al. (2019) and GNNs Joshi et al. (2019). Both supervised learning (SL) and reinforcement learning (RL) have been used for training with policy gradient based RL methods being the most popular Bello et al. (2017). Additionally, diffusion models have also been proposed as a technique for neural CO Sun & Yang (2023); Sanokowski et al. (2024). In addition to TSP, neural methods have been proposed to heuristically solve countless other NP-hard problems including graph matching Zanfir & Sminchisescu (2018), routing problems Zhou et al. (2025), maximum independent set Ahn et al. (2020) , and Boolean satisfiability Bünz & Lamm (2017) to name a few. In particular, there have been a number of neural approaches to Max-Cut and other closely related problems such as Maximum independent set (MIS) that are directly equivalent to the Ising problem studied in this work Dai et al. (2018); Karalias & Loukas (2021); Schuetz et al. (2022); Zhang et al. (2023); Sanokowski et al. (2023; 2024; 2025). Approaches include GNNs Schuetz et al. (2022) G-flow-nets, Zhang et al. (2023) and more recently diffusion samplers Sanokowski et al. (2024; 2025). For further references and recent reviews on neural CO we refer the reader to Cappart et al. (2022); I. Garmendia et al. (2024); Martins et al. (2025); Thinklab-SJTU (2021).

## 2.2 Dynamical System Approaches to Ising/Max-Cut (Ising Machines)

Over the years, there have been many physics-inspired approaches to solving the Max-Cut problem. These approaches are inspired by the fact that many physical systems naturally seek to minimize some quantity (e.g., physical energy) so if the problem objective can be mapped to this quantity then the physical device can sample good solutions to the desired optimization problem. These ideas date back to concepts like Hopfield neural networks Hopfield (1982), but have gained more attention again recently. In particular, there has been a lot of work showing that even for certain systems, simulating the physical dynamics numerically can lead to developing state-of-the art algorithms. These algorithms, typically called "Ising machines" often involve representing the solution of a Max-Cut/Ising problem as a set of continuous degrees of freedom which evolve dynamically over time. Some examples include coherent Ising machines (CIM) Wang et al. (2013); Yamamoto et al. (2017; 2020); Lu et al. (2023), analog iterative machines (AIM) Kalinin et al. (2023) which are inspired by photonics, oscillator Ising machines (OIM) Wang & Roychowdhury (2019) based on analog electronics as well as algorithms like simulated bifurcation machines (SBM) Goto et al. (2019), and chaotic amplitude control (CAC) Leleu et al. (2019; 2021); Leleu & Reifenstein (2025) which are inspired by more general physical principals. Although it has been demonstrated numerically that these algorithms are very effective at solving Max-Cut problems, it is not well understood why certain types of dynamics are more effective than others beyond some loose connections with the underlying physics. Additionally, as is common for heuristic algorithms for CO, there is often an amount of hyper-parameter tuning required for these algorithms to be effective on a certain class of problem instances.

## 2.3 Learning to Optimize and Algorithm Unrolling

Learning to Optimize (L2O) and algorithm unrolling are machine learning techniques commonly used for solving optimization problems where a simple iterative algorithm is typically used. The idea of algorithm unrolling is that instead of replacing the whole iterative algorithm with a many layered deep neural network, we modify the existing iterative algorithm to a more general version with more parameters (which is essentially a recurrent neural network) Monga et al. (2020); Chen et al. (2021); Kotary et al. (2023); Chen et al. (2024). These parameters can then be tuned by some machine learning technique so that we get an improved version of the original iterative approach. Because this technique requires many fewer parameters than the corresponding DNN parameterization, it is often much more efficient, scalable and interpretable Monga et al. (2020). Algorithm unrolling has traditionally focused on problems in sensing and signal reconstruction Chen et al. (2021); Balatsoukas-Stimming & Studer (2019); Gregor & LeCun (2010). A foundational example of this is in which ISTA (iterative shrinkage and thresholding algorithm), a commonly used algorithm for sparse signal construction, was extended to LISTA (learned ISTA). The convergence and reconstruction accuracy of ISTA were greatly improved by learning additional parameters Gregor & LeCun (2010). Although these example are mostly in settings where the underlying optimization problem is convex, there are some works that explore algorithm unrolling in the context of non-

convex optimization Tan et al. (2023); Wei et al. (2025); Song et al. (2024). However, to the best of our knowledge algorithm unrolling has not been explored for NP-hard combinatorial optimization with the exception of the integer linear programming (ILP) problem which is discussed in Chen et al. (2024).

### 2.4 Zeroth Order Optimization and Evolutionary Strategies

Most ML techniques require some sort of gradient estimator which usually based off of backpropagation. Examples of this are the typical stochastic gradient descent (SGD) used in supervised learning and the policy gradient method of reinforcement learning. On the other hand, zeroth-order methods do not use a backwards pass and compute the gradient using a finite-difference estimator. These types of methods have been proposed as alternatives to backpropagation and policy-gradient reinforcement learning, typically because of the smaller computational overhead required Salimans et al. (2017). In our case, we use a zeroth order method for a slightly different reason. Because the dynamics of an Ising machine are very complex and require many layers of computation, it is not possible to use backpropagation to estimate gradients accurately because of the vanishing/exploding gradient phenomenon. Similarly, if we try to use the policy gradient method on an Ising machine an equivalent problem arises. Because there are so many small steps (decisions) that an Ising machine makes in a single trajectory, when using the policy gradient method we get a very noisy gradient single because it is hard to accurately attribute each of these decisions to the success of the algorithm. For more details and numerical results see appendix E. Note that many previous works such as Zhang et al. (2023) and Sanokowski et al. (2024) attempt to fix this problem of reward attribution for CO algorithms, however we take a different approach and use an entirely different optimization technique which is not based on the REINFORCE algorithm Williams (1992).

### 2.5 Our Contribution

In this work, we propose a method of neural CO which essentially applies the idea of algorithm unrolling to dynamical Ising machines. For our training method, we use a zeroth order optimization method Reifenstein et al. (2024) instead of the more typical backpropagation or policy-gradient based methods. We parameterize the update step of an Ising machine with a neural network allowing it to learn optimal search dynamics in a data-driven way. To the best of our knowledge, our method is novel in the following ways:

- We apply the techniques of algorithm unrolling to the NP-hard Max-Cut problem.
- We use zeroth-order optimization to tune a neural network in the context of combinatorial optimization.
- We show that effective dynamics for Ising machines can be learned from scratch in a data-driven way.

## 3 Proposed Method

### 3.1 The Ising Problem

In this work, we will consider the NP-hard Ising problem defined as follows. The goal is to minimize a quadratic objective function with respect to a set of binary $N$ variables:

$$\text{minimize} \quad \sum_{i,j} J_{ij}\sigma_i\sigma_j - \sum_i l_i\sigma_i \quad \text{subject to} \quad \sigma \in \{-1, 1\}^N \tag{1}$$

where the problem instance is specified by the symmetric $N$x$N$ matrix $J$ and $N$ dimensional vector $l$. This problem can be shown to be mathematically equivalent to several other optimization problems including Max Cut (MCut), Max Clique (MC), Max Independent Set (MIS) and QUBO (Quadratic Unconstrained Binary Optimization) problems (see appendix A). In this work we will often use the term "Ising problem" to refer to this class of problems which are described by optimizing a set of binary variables over a quadratic objective function.

Figure 1: a) Diagram showing the flow of information in a neural network Ising machine. The coupling fields (purple), calculated by aggregating the influence of the $N-1$ other variables, are saved from previous iterations. They are then fed into the neural network model and used to decide the new spin variable which is then used to compute the next coupling field. See sections 3.2 and 3.3 for a concrete mathematical description. b) High level overview of out method relative to other approaches to CO, inspired by figure 15 of Monga et al. (2020). c) Cartoon depiction of the zeroth-order evolutionary optimization algorithm we use based off of Reifenstein et al. (2024). A distribution of parameters (represented by red circles) is evolved over many iterations to close in on an optimal parameter configuration (blue dot).

## 3.2 Ising Machines

An Ising machine is most generally defined as a dynamical system which is used to optimize the Ising problem. However, in this work we will give it a specific mathematical definition as

$$x_i(t) = F(t, h_i(0), ..., h_i(t-1)) \tag{2}$$

$$h_i(t) = \sum_j J_{ij} x_j(t) + \frac{1}{2} l_i. \tag{3}$$

The Ising machine dynamics are fully determined by the function $F$. The variables $x_i(t)$ denote the current approximate solution of the Ising problem (although they can have continuous values) and the variables $h_i(t)$ can be interpreted as a discrete gradient of the objective function with respect to the current approximate solution. In addition, $F$ may have some stochastic component to it. The algorithm is carried out by starting with $x_i(0)$ at some random value (determined by $F$) and then iterating the above equations by $T$ steps. At each step, a possible solution can be calculated as $\sigma_i(t) = \text{sign}(x_i(t))$ and typically we take the best solution over the course of the trajectory to be the output. Because of the stochastic nature, in practice many trajectories are often computed and the best solution out of all of them is used. For specific examples of dynamical Ising machines and how they map to this formulation, refer to appendix B.

## 3.3 MLP Parameterization

The key concept behind our method is to parameterized the function $F$ with a simple multilayer perceptron (MLP) neural network. To do this, we restrict the history of $h_i$ variables used to a specific length $T_c$ and use these as the input layer of our network. For this work we use a two layer network with $\tanh$ activation functions. Additionally, we do not include bias parameters. This is because in order for the algorithm to respect the symmetry of the Ising problem we want the resulting function to be odd with respect to every input. We can express this function explicitly as follows:

$$F(t, h(0), ..., h(t-1)) = \text{MLP}(t, h(t-T_c), ..., h(t-1)) = \tag{4}$$

$$\tanh\left[W^0(t)\eta + \sum_{k \in \{0,...,D-1\}} W^1_{1,k}(t) f_{\text{nl}}\left(\sum_{s \in \{0,...,T_c-1\}} W^2_{k,s}(t) h(t-T_c+s)\right)\right] \tag{5}$$

where $D$ denotes the number of hidden neurons, $\eta$ is $\mathcal{N}(0,1)$ gaussian noise, and $f_{\text{nl}}$ is the nonlinear activation function $f_{\text{nl}}(x) = x + \tanh(x)$. $W^0(t)$,$W^1(t)$ and $W^2(t)$ are the tunable weight matrices of dimensions 1x1, 1xD and $DxT_c$ respectively. Note that the weight matrices are indicated to have dependence on $t$. This is because we want the Ising machine's dynamics to be allowed to vary over the course of the trajectory which is something that is important to many Ising machines. To

complete this parameterization, we first flatten the weight matrices into one vector $\theta(t)$ of dimension $1 + D + DT_c$. Then we introduce another hyperparameter $M$ which corresponds to the number of degrees of freedom for which each parameter can vary with respect to time. More specifically, we can express $\theta_i(t)$ in terms of a $(1 + D + DT_c)$x$M$ matrix $\Theta_{i,m}$ as follows:

$$\theta_i(t) = \sum_{m \in \{0,...,M-1\}} \Theta_{i,m} f_m(t/T) \tag{6}$$

where $f_m$ are a set of functions on the interval $[0, 1]$ which can be used as a basis to describe a general smooth function. In our work we use the Fourier basis described by

$$f_m(\tau) = \begin{cases} \cos(\frac{m}{2}\pi\tau) & m \in \text{even} \\ \sin(\frac{m+1}{2}\pi\tau) & m \in \text{odd} \end{cases} . \tag{7}$$

Together, this gives a total of $(1 + D + T_c D)M$ total parameters. We will refer to the algorithm created by iterating these equations as "neural network parameterized Ising machine" (NPIM). In addition to the network defined in equation 5, we also consider another version in which the outer $\tanh$ nonlinearity is replaced by the discontinuous $\text{sign}$ function causing the $x_i(t)$ variables to be binary. We will refer to the resulting algorithms as cNPIM and dNPIM respectively (corresponding to continuous and discrete coupling). Although dNPIM is technically a special case of cNPIM (by scaling the weights), dNPIM tends to have different inductive biases and better generalization as shown in section (see section 4.5). We find that the specific choice of temporal basis described in eq 7 (Fourier, Chebyshev, Legendre) has only a minor effect on performance, while the dominant factor is the number of temporal modes $M$ available, as observed in Appendix C.2 (Fig. 5).

### 3.4 Parameter Tuning and Reward Function

To optimize the parameters of our model we use a zeroth-order evolutionary optimization method based off of Reifenstein et al. (2024). To do this, we choose a reward function which incentivizes trajectories which are successful at finding good values of the objective function. Depending on our goal (i.e. which benchmark we are training for) we use one of two reward functions which are described in appendix F. The optimization process can then be formalized as follows. A distribution in the space of network weights $\theta \in \mathbb{R}^P$ (where $P = (1 + D + T_c D)M$) is described by two variables $\theta_x \in \mathbb{R}^P$ and $\theta_L \in \mathbb{R}^{P \times P}$. Then, we define the reward function $\rho$ which takes as an in a trajectory of an NPIM and outputs a real number (see section F for specific definitions). Our goal is to maximize the expected reward function which can be written as

$$\mathcal{R}(\theta_x, \theta_L) = \mathbb{E}_{v,\eta,J} \, \rho(\text{traj}(\theta_x + \theta_L v, \eta, J)) \tag{8}$$

where the expected value is taken over three distributions. $v$ is a random variable in $N(0,1)^P$ which is mapped to a perturbation in the parameter space by the matrix $\theta_L$. $\eta$ is a random vector corresponding to the stochastic behavior of the trajectory dynamics themselves, and lastly $J$ is an instance of the Ising problem chosen from the relevant distribution. The notation $\text{traj}(\theta, \eta, J)$ is meant to symbolize a trajectory of the NPIM dynamics with the given network weights $\theta$, instantiation of noise $\eta$ and problem instance $J$. Following the equations in Reifenstein et al. (2024) we then estimate the gradient of $\mathcal{R}$ with respect to both $\theta_x$ and $\theta_L$ by computing samples from this distribution. At each step, the estimated gradients are then used to update both $\theta_x$ and $\theta_L$. A single update of both $\theta_x$ and $\theta_L$ we will refer to as an "epoch" in this work. For more details including equations and hyper-parameters, see appendix G

## 4 Analysis of Learned Dynamics

### 4.1 Example of learned dynamics: Emergence of Momentum in Single Layer Network

In order to illustrate the relationship between network weights, Ising machine dynamics, and algorithm performance we will briefly consider a simplified example in which we have a single layer network with fixed weights ($M = 1$) and 10 input neurons ($T_c = 10$). In figure 2 we show how this network evolves over the course of the training process. In the first few epochs the network quickly learns a greedy "steepest descent" strategy. This is reflected by all of the network weights being

negative (bottom middle of figure 2). However, because set of Ising problem instances used are non-convex, this basic strategy causes the machine to often get trapped in solutions which are not globally optimal. Thus, during the training process the network weights gradually become modified to allow for a more effective search procedure that includes some additional "momentum" effect that kicks it out of these meta-stable state. This is depicted in the upper and lower right plots where some of the network weights become positive (red).

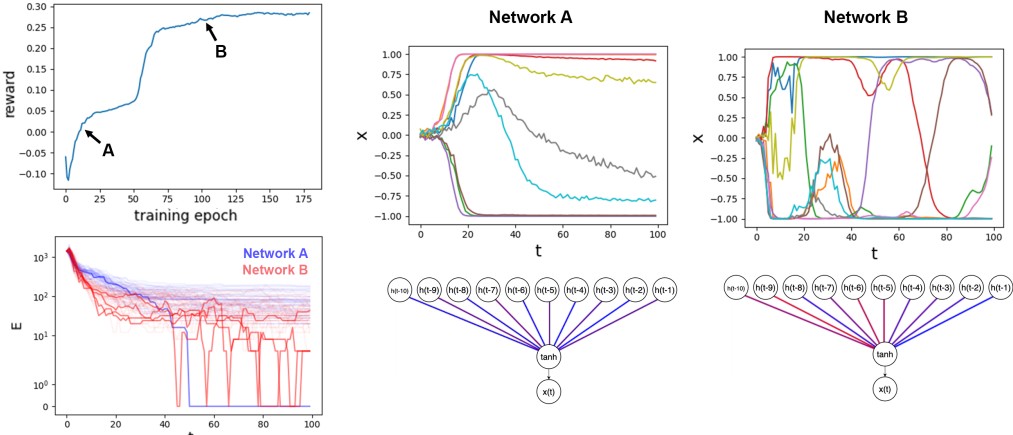

Figure 2: Example of training single layer neural network Ising machine. Upper left: the average reward (success rate) of the network with respect to training epoch. The reward starts negative because of an initial bootstrapping phase in which bad trajectories are penalized. Two snapshots of the network parameters are taken and shown in the right two figures: network A at epoch 19, and network B at epoch 99. Lower left: residual Ising energy (difference with best known solution) is shown as a function of iteration step for both networks. Darker colored trajectories indicate the ground state was found. Bottom middle and right: network weights of network A and network B respectively. Blue and red connections depict negative and positive network weights respectively. Top middle and right: trajectory of $x_i(t)$ variables for network A and network B respectively. Each color represents a different variable of the Ising problem.

## 4.2 EFFECT OF ARCHITECTURE ON PERFORMANCE

As shown in section 4.1, a simple single layer network with fixed weights can be effective at learning the complex dynamics required of solving these optimization problems. This raises the question of how important a more complicated multi-layer network is, and to what extent parameter modulation (annealing) is necessary for the algorithm to be effective. However, based on our experimentation with different network architectures it appears that both increasing the number of hidden neurons and degrees of freedom for the annealing schedule improve algorithm performance. In figure 3c and table 3 we show the success rate of both cNPIM and dNPIM on $N = 100$ SK problem instances for different network configurations. We see a clear trend in which a greater number of parameters results in improved performance, although there may be a saturation around 50 parameters, the results indicate that the network is learning some non-trivial strategy that needs many parameters to describe. Interestingly, as long as the number of parameters is large, the exact type of parameters (i.e. tradeoff between $T_c$, $D$ and $M$) doesn't seem to have a large effect on performance. For more details, single-parameter sensitivity sweeps over $T_c$, $D$, and $M$ are provided in Appendix C.1 (Fig. 4).

## 4.3 BOOTSTRAPPING AND FINE TUNING

In order to train the network on hard problem instances, it is often not sufficient to simply start with random parameters. This is because the success rate of finding the ground state will be zero or close to zero so there will be no gradient signal for the optimizer to use. To fix this problem we use various forms of bootstrapping and fine-tuning, in which the network is first tuned on an

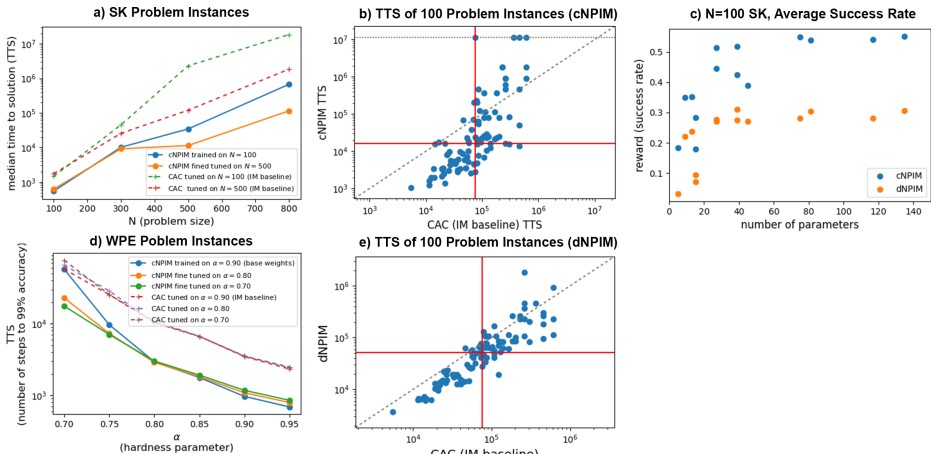

Figure 3: a) Performance (Time to solution) of cNPIM on Sherrington-Kirkpatrick (SK) problem instances. Colored traces show performance with and without fine-tuning showing limited (but nonzero) ability to generalize over problem size. Dotted trace shows baseline Ising machine performance (Chaotic amplitude control Leleu et al. (2019; 2021); Leleu & Reifenstein (2025)). b, e) Scatter plot showing the TTS of 100 random SK problem instances of problem size $N = 800$ against that of CAC for cNPIM and dNPIM respectively. c) Success rate for different architectures of cNPIM and dNPIM as a function of total parameter count. The same data is shown in table 3. d) TTS is shown as a function of hardness parameter for the Wishart planted ensemble (WPE) problem instances Hamze et al. (2020). Colored traces show cNPIM fine-tuned on different hardness parameters while dotted line shows Ising machine baseline (CAC).

easier version of the problem and then fine-tuned on the desired instance distribution. In figures 3a and 3d we show two examples of bootstrapping and fine-tuning. For example, in figure 3a the network is first trained from scratch (random initialization) on SK problem instances of problem size $N = 100$. Then, this pretrained network is fine-tuned on problem size $N = 500$. Performance of both networks is shown in blue and orange traces respectively, showing that the fine-tuned network is more effective especially for larger problem sizes. This process is necessary because training a network from scratch at the larger problem size ($N = 500$) is not possible. For more details on training process see appendix F.

## 4.4 Out of Distribution Performance

In figures 3a and 3d we show the performance of cNPIM with respect to problem size and instance hardness parameter respectively. In both cases, when the network is tuned on a specific problem distribution, the fine-tuned weights are still successful at solving problems in different (but closely related) distributions. However, performance tends to degrade the more the distribution differs from the one it is tuned on as expected. This shows that although some out-of-distribution generalization is possible, fine-tuning is still important in order to get the desired performance.

## 4.5 Overfitting and Differences between cNPIM and dNPIM

In figures 3b and 3e we show the instance-wise performance of cNPIM and dNPIM respectively against that of the chaotic amplitude control (CAC) algorithm Leleu et al. (2019; 2021); Leleu & Reifenstein (2025). Because the network is trained to optimize average success rate over all instances, this can result in overfitting in which the success rate of some problem instances is very large whereas others will have zero or very low success rate. This is depicted in figure 3b where many of the easier instances have low TTS (high success rate) for cNPIM compared to CAC whereas some hard instances were not solved at all by cNPIM indicated by their placement on the horizontal dotted line. On the other hand, in figure 3e we see this effect is much less prevalent for dNPIM. Although cNPIM achieves a larger reward value (average success rate), and smaller TTS for the median difficulty problem instances (indicated by red lines), it struggles on the hardest

problem instances relative to dNPIM and CAC.

Although this phenomenon is not fully understood, we believe that because cNPIM uses continuous coupling it learns to optimize some relaxed version of the underlying discrete Ising problem. Although this relaxed problem may align well with the real problem for some instances, it doesn't for others, making it unreliable if we want to find the true ground state. On the other hand, because dNPIM uses discrete couplings the internal state of the algorithm is always based on true solutions to the underlying Ising problem. This forces it to be more faithful in its ability to search over the true solution space which may cause it to take longer for the easier problem instances.

# 5 BENCHMARK RESULTS

Because our method is closely related to both literature from the machine learning community on neural CO as well as the literature on dynamical Ising machines, to benchmark our algorithm we will include common benchmarks from both fields. Each field differs in what type of problems and what performance metric is used.

In the literature on neural CO, typically both average objective value and computation time are reported Zhang et al. (2023); Sanokowski et al. (2025). Additionally, common benchmark problems include maximum independent set and Max-Clique problems based off of graphs from Xu et al. (2005) and Max-Cut problems from the Barabási–Albert (BA) distribution Albert & Barabási (2002). In table 1 we compare against the results of Sanokowski et al. (2025) on MIS, Max-Clique and Max-Cut problems. Although Sanokowski et al. (2025) also includes results on the maximum dominating set problem, we omit these because it is not directly mappable to the quadratic Ising problem. However our framework can easily be extended to other types of problems like this (see appendix D) which can be explored in future works. We find that in four out of the five cases dNPIM is able to achieve a better average objective value than the results of Sanokowski et al. (2025). However, in the case of the larger graphs our method does take longer. Although we are using the same hardware as Sanokowski et al. (2025) this difference could have something to do with the sparse graph library used for the results in Sanokowski et al. (2025) as opposed to the dense PyTorch matrix-matrix product used in our implementation. So without further optimization it is unclear if this difference in speed is inherent to the algorithm or the implementation.

In literature on Ising machines, time to solution (TTS) is typically used as a metric. TTS takes into account both computation time of a single run of the algorithm and the quality of solutions achieved per run into a single metric. TTS is defined as an estimate of the amount of time you would need to run the algorithm to have a 99% chance of finding the solution. Because these are NP-hard problems an we don't know the true optimum we use "solution" to mean the best solution found by the algorithms we are benchmarking. For more details on TTS and how it is calculated see appendix H. For benchmark problem instance we use the famous G-set instance which are a set of both weighted an unweighted graphs with a variety of structures. These graphs are typically interpreted as Max-Cut problems for benchmarking. In order to train our network, for each type of graph in the G-set we generate a training set of problem instances which is used to fine-tune a network for that specific set of graph parameters (see appendix I for details). We compare the resulting algorithm against the results of Reifenstein et al. (2021) and Goto et al. (2021). We use the cut values reported in these works when computing TTS. Note that for the results of Reifenstein et al. (2021) and Goto et al. (2021) algorithm parameters are also tuned for each instance type. We find that on almost all problem instances dNPIM outperforms the existing Ising machine state-of-the art with the exception of the unweighted planar instances. These instances are more difficult and other Ising machine algorithms struggle on them as well, especially dSBM (as shown in Reifenstein et al. (2021)). We believe that with more careful optimization and improvements to the architecture our method could achieve SOTA performance on all G-set instance but we leave this in-depth exploration for future works.

Overall, we find that in almost all cases we have explored, our NPIM approach is able to compete with state-of-the art results. This is promising because the simplicity and flexibility of the method makes it attractive as a technique that can quickly be adapted to a wide variety of optimization problems.

| Method | MIS-small | | MIS-large | | MaxCl-small | | MaxCut-small | | MaxCut-large | |
|---|---|---|---|---|---|---|---|---|---|---|
| | Size ↑ | time ↓ | Size ↑ | time ↓ | Size ↑ | time ↓ | Size ↑ | time ↓ | Size ↑ | time ↓ |
| Gurobi | $20.13 \pm 0.03$ | 6:29 | $42.51 \pm 0.06^*$ | 14:19:23 | $19.06 \pm 0.03$ | 11:00 | $730.87 \pm 2.35^*$ | 17:00:00 | $2944.38 \pm 0.86^*$ | 2:35:10:00 |
| LTFT (r) | 19.18 | 1:04 | 37.48 | 8:44 | 16.24 | 1:24 | 704 | 5:54 | 2864 | 42:40 |
| DiffUCO | $19.42 \pm 0.03$ | 0:02 | $39.44 \pm 0.12$ | 0:03 | $17.40 \pm 0.02$ | 0:02 | $731.30 \pm 0.75$ | 0:02 | $2974.60 \pm 7.73$ | 0:02 |
| SDDS: $rKL$ w/ $RL$ | $19.62 \pm 0.01$ | 0:02 | $39.97 \pm 0.08$ | 0:03 | $\mathbf{18.89 \pm 0.04}$ | 0:02 | $731.93 \pm 0.74$ | 0:02 | $2971.62 \pm 8.15$ | 0:02 |
| dNPIM (top 30) | **19.9** | 0:02 | **40.297** | 1:20 | 18.7 | 0:02 | **734.908** | 0:02 | **2988.551** | 1:20 |

Table 1: Comparison of different methods on Max Independant Set (MIS), Max Clique (MaxCl) and MaxCut problems. Solution size (higher is better) and computation time (lower is better) are used as dual performance indicators. We compare with data from Sanokowski et al. (2025) which includes benchmark results of DiffUCO Sanokowski et al. (2024) and LTFT Zhang et al. (2023) as well. Computation times are based on PyTorch code running on and NVIDIA A100 GPU. "top 30" refers to the fact that since our algorithm is less computationally intensive per trajectory than the other algorithms we compare it to we run it 30 times in parallel and then use the best solution found out of these trajectories for our comparison.

| Method | N=800, R, + TTS ↓ | N=800, R, +/- TTS ↓ | N=800, T, +/- TTS ↓ | N=800, P, + TTS ↓ | N=800, P, +/- TTS ↓ |
|---|---|---|---|---|---|
| CAC | 2.09e+05 | 4.31e+05 | 3.38e+05 | **1.81e+06** | 8.87e+05 |
| CFC | 2.39e+05 | 2.24e+05 | 2.22e+05 | 2.00e+06 | 3.44e+05 |
| dSBM | 4.00e+05 | 3.59e+05 | 4.08e+05 | 2.12e+07 | 5.25e+06 |
| dNPIM | **1.00e+05** | **6.55e+04** | **5.51e+04** | 4.42e+07 | **2.04e+05** |

Table 2: Comparison of different methods on the G-set max-cut problem instances. Time-to-solution is used as performance metric. Time-to-solution is reported in units of number of iterations to solution. This is because the compute intensive matrix vector product is the computational bottleneck for each algorithm. In this table, we report medians over each group of instances, but for instance-wise performance see table 4. Target cut values used to evaluate the success of the algorithm are taken from Goto et al. (2021) and represent the current best known cut values for these instances. State of the art Ising machine TTS is obtained by taking the best TTS from Reifenstein et al. (2021) which includes the results of Goto et al. (2021) as well.

## 6 Conclusions and Discussion

We have presented a novel data-driven method for solving combinatorial optimization problems. We use ideas from algorithm unrolling, Ising machines and zeroth-order optimization in a new way to learn algorithms that can achieve state-of-the art performance on commonly used benchmarks. In addition to being novel, the simplicity of our approach makes it (in principle) easily generalizable to many types of problem instances. To conclude, we will discuss some current limitations of our approach and future directions that should be explored.

In the context of our work there are two types of scalability: with problem size ($N$), and with number of network parameters. We believe that our method achieves good scaling with respect to problem size relative to the general difficulty of scaling in CO (see figure 3a). However, scaling with number of parameters can be a potential limitation. This stems from the fact that we use a zeroth-order optimization method which will cause an additional overhead in the optimization when more parameters are added (for example see figure 4). This may limit the networks capability to learn more sophisticated dynamics (i.e. non-local moves) which maybe required to solve certain types of problems. An interesting future direction would be to combine the zeroth-order method used in this work with some sort of policy gradient or backpropagation-like method to see if the network could scale to a larger number of parameters.

Another limitation of our method is the problem of explainability. Although this problem is common in ML approaches in general and, to a lesser extent, dynamical Ising machines, we have not contributed much in this work to fix this explainability issue. The best we can do currently is draw connections to physical concepts used in the optimization literature such as "momentum" and "annealing". We show to some extent that these phenomena are emergent properties of our network when it is trained with the sole objective of maximizing reward (see figure 2). However, this does not answer the question of why these dynamics are so important for certain problem instances. A more detailed understanding of the dynamical complexity generated by the learned iterative map is still needed, and is an interesting direction for future works.

Although we have tested our approach on a variety of benchmarks, these problem instances are synthetic and are constrained to the class of quadratic optimization over binary variables. To further study our method it will be necessary to test it on different types of CO problems such as SAT, integer programming and TSP (see section D) and also consider problems of industrial or academic interest. Because of the simplicity and flexibility of our method, we believe it is likely that our approach can be adapted provide an efficient solution in some real-world applications.

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

# A  EQUIVALENCE OF MAX-CUT, MAX-CLIQUE, MIS AND QUBO TO THE QUADRATIC ISING PROBLEM

In this section we will show the exact mathematical form between a these different types of combinatorial optimization problems. The graph-based problems will be described by a adjacency matrix $A$ while the QUBO coupling matrix will be denoted $Q$.

## A.1  MAX-CUT

$$J_{ij} = A_{ij} \quad l_i = 0 \tag{9}$$

## A.2  MAX-CLIQUE

$$J_{ij} = (1 - A_{ij}) \quad l_i = 0.9 + \sum_j (1 - A_{ij}) \tag{10}$$

## A.3  MIS

$$J_{ij} = A_{ij} \quad l_i = 0.9 + \sum_j A_{ij} \tag{11}$$

## A.4  QUBO

$$J_{ij} = Q_{ij} \quad l_i = \sum_j Q_{ij} \tag{12}$$

# B  ADDITIONAL DETAILS ON ISING MACHINES

In this section we will show some of the equations for other Ising machines and how they fit into the mathematical framework of equations equation 2 and equation 3.

## B.1  CHAOTIC AMPLITUDE CONTROL (CAC)

Chaotic amplitude control Leleu et al. (2019; 2021) is described by the following iterative update equations:

$$x_i(t+1) = x_i(t) + \mathrm{dt}\left(-ax_i(t) - x_i(t)^3 - \xi e_i(t)\left(\sum_j J_{ij}x_j(t) + l_i\right)\right) \tag{13}$$

$$e_i(t+1) = e_i(t) + \mathrm{dt}\beta e_i(t)\left(1 - x_i(t)^2\right) \tag{14}$$

With $x_i(0) \in \mathcal{N}(0,1)$ and $e_i(0) = 1$. This can then be put into the form of equations 2 and 3 by defining $F$ recursively as

$$x(t) = F(t, h(0), ..., h(t-1)) \tag{15}$$

$$x(t+1) = x(t) + \mathrm{dt}\left(-ax(t) - x(t)^3 - e(t)h(t)\right) \tag{16}$$

$$e(t+1) = e(t) + \mathrm{dt}\beta e(t)\left(1 - x(t)^2\right) \tag{17}$$

$$x(0) \in \mathcal{N}(0,1) \quad e(0) = 1 \tag{18}$$

similarly, other Ising machines such as CIM Wang et al. (2013), and SBM Goto et al. (2021) can be described in a recursive way like this.

## B.2 Analog Iterative Machine (AIM)

We will also include the equations for the analog iterative machine Kalinin et al. (2023) because it is an interesting case in which the forumla for $F$ can be written explicitly. An AIM is described by

$$z_i(t+1) = z_i(t) + \text{dt}\left(-\alpha \sum_j J_{ij} \tanh(z_j(t)) - \beta(t)z_i(t) + \gamma(z_i(t) - z_i(t-1))\right) \quad (19)$$

where $z_i(0)$ and $z_i(1)$ are initialized randomly. If we let $\beta(t) = \beta$ to be constant, then we can write

$$z_i(t+1) = \sum_{t'=0,...,t+1} -\frac{\lambda_1^{t-t'} - \lambda_2^{t-t'}}{\lambda_1 - \lambda_2}\alpha \sum_j J_{ij} \tanh(z_j(t')) \quad (20)$$

where $\lambda_1$ and $\lambda_2$ are eigenvalues of the matrix $\begin{pmatrix} 1 + \text{dt}(-\beta + \gamma) & \text{dt}\gamma \\ 1 & 0 \end{pmatrix}$. This allows us to write $F$ in the explicit form

$$\tanh(z(t)) = F(t, h(0), ..., h(t-1)) = \tanh\left[\sum_{t'=0,...,t+1} \frac{\lambda_1^{t-t'} - \lambda_2^{t-t'}}{\lambda_1 - \lambda_2}h(t')\right] \quad (21)$$

In addition to being explicit, this mathematical form is also equivalent to a single layer cNPIM with $T_c = \infty$ (or just $T_c \geq T$).

## C Effect of Architectural Hyperparameters

### C.1 Hyperparameter sweep

We sweep one hyperparameter at a time to isolate its effect on the performance of neural parameterized Ising machines. Figure 4 shows results for varying the history length $T_c$ (panel a), the number of hidden neurons $D$ (panel b), and the number of Fourier modes $M$ controlling the time dependence of parameters (panel c). In each case, we compare the continuous (cNPIM) and discrete (dNPIM) variants trained on $N = 100$ Sherrington–Kirkpatrick instances uszing the success-rate reward. The optimization procedure uses $R = 400$ trajectories per epoch, batch size $B = 20$, and was run for 800 epochs.

In conclusion, all three architectural parameters materially influence performance, with larger $T_c$, $D$, and $M$ generally improving the success rate. The small decrease observed at the largest values is most likely due to the limited number of training epochs, which prevents full convergence of the higher-capacity models rather than indicating a true decline in effectiveness.

### C.2 Choice of temporal basis

We compare different temporal basis functions used to parameterize the time dependence of NPIM weights. Figure 5 shows results for Fourier, Legendre, and Chebyshev bases, evaluated for $M \in 1, 3, 5$ with fixed history length $T_c = 8$ and hidden dimension $D = 3$. Both cNPIM and dNPIM are trained on $N = 100$ Sherrington–Kirkpatrick instances using the success-rate reward. Training was run for 400 epochs with $R = 400$ trajectories per epoch and batch size $B = 20$. The results indicate that all three bases are viable choices for encoding temporal variation, with performance improving as $M$ increases regardless of basis type. Differences between bases are relatively minor at small $M$, and all yield comparable performance at larger $M$, suggesting that the precise functional form of the temporal basis is less critical than the number of degrees of freedom provided.

## D Generalized Algorithm

In this section we will show one way in which the proposed framework can be generalized to combinatorial and other types of optimization problems beyond the Ising problem. Imagine a general setting where we are given $N$ variables which are chosen from a set $S$. An objective function is

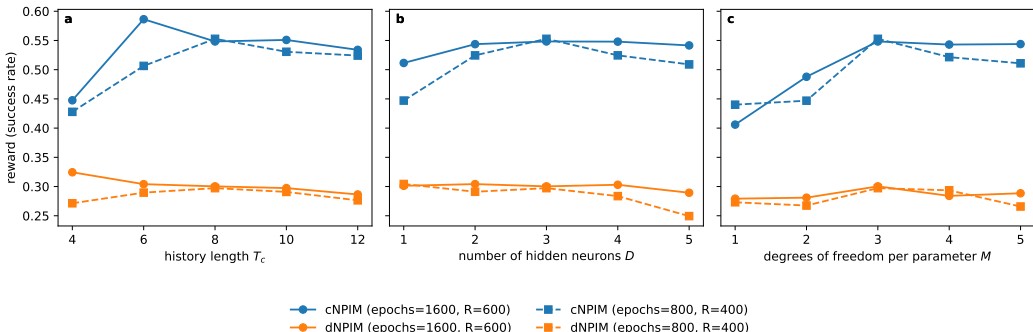

Figure 4: Hyperparameter sweeps of NPIM on SK instances. a) Final success rate as a function of history length $T_c$ with $D = 3$ and $M = 3$. b) Success rate as a function of hidden neurons $D$ with $T_c = 8$ and $M = 3$. c) Success rate as a function of Fourier modes $M$ with $T_c = 8$ and $D = 3$. Curves compare the continuous (cNPIM) and discrete (dNPIM) variants trained on $N = 100$ SK instances. Batch size $B = 20$, and the success-rate reward.

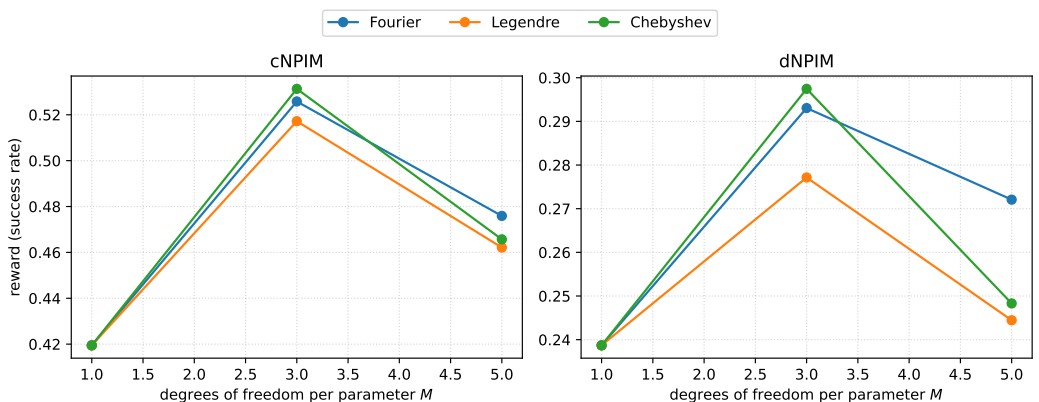

Figure 5: Comparison of temporal basis functions in NPIM. Final success rate for Fourier, Legendre, and Chebyshev bases as a function of degrees of freedom per parameter $M$, with $T_c = 8$ and $D = 3$ fixed. Left: cNPIM. Right: dNPIM. Networks are trained on $N = 100$ SK instances using the success-rate reward, with 400 epochs, $R = 400$ trajectories per epoch, and batch size $B = 20$. Performance improves as $M$ increases, and differences between bases are small once sufficient degrees of freedom are available.

defined as $g : S^N \to \mathbb{R}$ and additionally a "gradient direction" operator $\partial g : S^N \to S^N$ which points in a direction of increased objective value. Additionally, we define a "gradient magnitude" operator $\Delta g : S^N \to \mathbb{R}^N$ to estimate the change in objective value cause by each individual variable update. We then define an iterative algorithm

$$x_i(t+1) = F\left(t,\, x_i(t), \partial g_i(t), \Delta g_i(t),\, x_i(t-1), \partial g_i(t-1), \Delta g_i(t-1),\, ...., \, x_i(0), \partial g_i(0), \Delta g_i(0)\right) \tag{22}$$

the function $F$ can then be parameterized by some sort of neural network depending on the exact form of $S$. This framework is meant to be general for many types of optimization problems but in many specific examples it can be made much simpler. For example, we can consider a more general optimization over a set of binary variables where $S = \{-1, +1\}$ like Ising, but the objective function takes some more general form. Then, $\Delta g_i(x) = g(x\mid_{x_i=+1}) - g(x\mid_{x_i=-1})$, $\partial g_i(x) = \text{sign}(\Delta g_i(x))$. This includes problems like boolean SAT or MDS (maximum dominating set). Additionally, we can extend this framework to problems like integer programming problems where $S = \mathbb{Z}$ or some subset of $\mathbb{Z}$ as well as continuous optimization where $S = \mathbb{R}$. In these cases $\Delta g$ and $\partial g$ would represent the discrete and continuous gradients, respectively.

However, when it comes to problems like TSP or other routing-like problems it may be difficult to apply this framework. This is because it is unclear how to "factor" the space of solutions into a product of $N$ copies of a set $S$. This also has to do with the fact that for problems like TSP, more non-local updates might be needed such as what is used in heuristics like 2-opt, 3-opt and Lin-Kerrington. This could represent a general drawback of this type of framework which might also be reflected in its poorer performance on max-clique problems (see table 1). However we are optimistic that this drawback could potentially be overcome by a more sophisticated architecture that allows for non-local updates.

## E  ZEROTH-ORDER OPTIMIZER VS POLICY GRADIENT METHOD

Although it has not been touched on much in the main text, a key result of our findings is that training Ising machine dynamics using the policy gradient method of RL does not appear to be very effective. In this section we will provide some more details and briefly explain why we believe this is.

To formalize this we will first need to write the Ising machine dynamics in the form of a Markov decision process so we can apply the policy gradient. The set of possible states in this case will be $x \in \{-1, +1\}^N$ and each step the algorithm will output a probability distribution over this set.

$$P(x = \sigma) = \prod \frac{1 + \sigma_i F(t, h_i(0), ..., h_i(t-1))}{2} \tag{23}$$

Using this formulation we can the apply the policy gradient method Williams (1992) to tune the network parameters as well as a zeroth-order method. This lets us directly compare the two optimization approaches. We find, as show in figure 6, that the zeroth-order method is much more efficient and finds good parameters more quickly. Additionally, this discrepancy is more prominent when the problem size ($N$) is increased (not shown in figure). It is for this reason that this work is solely focused on using a zeroth-order method to optimize the parameters and do not consider other types of gradient estimators.

To understand more concretely why these methods differ in efficacy we can look at how the different gradient estimators work. As mentioned in the main text in section 2.4 we believe the failure of the policy-gradient method has to do with the fact that, for larger problem sizes, there are essentially many more "decisions" that the Ising machine has to make. Because of this, each decision on average contributes less to the success of the algorithm. So, using a gradient estimator at the level of a single decision is going to result in a very noisy estimate. More concretely, a MDP-based Ising machine will make a total of $NT$ decisions over to course of a $T$-step trajectory of problem size $N$. If we make the simplifying assumption that each decision contributes $\sim \frac{1}{NT}$ to the total success of the algorithm (i.e. choosing one sign for the spin variable will result in a roughly $\frac{1}{NT}$ larger probability of success), then this results in the gradient estimate for a single decision to have an SNR of roughly $O((NT)^{-1})$. Even once we average over $NT$ total decisions we still have an unfavorable SNR scaling of $O((NT)^{-\frac{1}{2}})$.

On the other hand, making estimate of SNR for zeroth-order methods is not dependent on the number of "decisions" that the algorithm makes, but more so the reward landscape itself. One way of understanding this is that, whereas the policy gradient method relies on perturbations caused by the randomness of the decisions, the perturbations in the parameter space causes a sort of "correlated perturbation" over all $NT$ decisions simultaneously which greatly increases the SNR of the estimator.

Although this mathematical intuition can be useful, the exact reason for which the policy gradient method fails in this case is not well understood at the moment, and potentially could be a focus of future works. Currently, we have come to this conclusion primarily based on extensive trial and error, most of which is not included in this text.

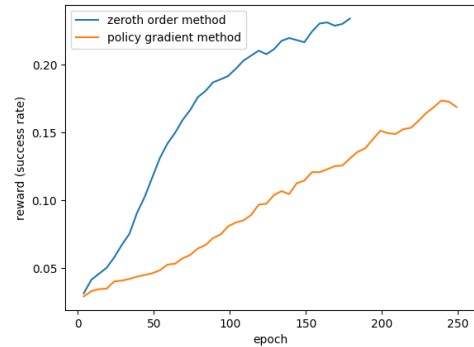

Figure 6: Training reward of a zeroth-order optimization method and a policy gradient based method on the same instance distribution and network architecture.

# F  REWARD FUNCTIONS

In this work we use two reward functions for training. All of the reward functions used are a function of $E_{\text{opt}}$, the best Ising energy in the given trajectory and $E_0$ the best energy found by all previous runs of all algorithms (the supposed "ground energy"). Since we don't know the true value for $E_0$ in practice, during training we keep track of the best energy found by the algorithm over all previous trajectories and use this as $E_0$. Since each instance is seen many times during training, towards the end of the training this value has stabilized and the algorithm reaches this energy with relatively high probability. This, along with the fact that it is the same energy found by other algorithms we tested makes us postulate that by the end of the training in most cases this is the ground state energy. This is also the logic behind why we use time to solution (TTS) as a metric. In cases where we are using TTS as an metric, then we wish to optimize the success rate and use the reward function

$$\mathcal{R}_{\text{succ}}(E_{\text{opt}}) = \begin{cases} 1 & \text{if } E_{\text{opt}} = E_0 \\ -\frac{1}{2} & \text{if } E_{\text{opt}} >= \frac{1}{2}E_0 \\ 0 & \text{otherwise} \end{cases} \tag{24}$$

the purpose of the middle case is to penalize really bad trajectories. This is important during the beginning of training to get a good reward signal when there might not be many successful trajectories, but during the end of the training there are no longer any of these bad trajectories so the reward landscape that is ultimately being optimized is equivalent to success rate. This two layered reward function serves a similar purpose to the bootstrapping and fine-tuning described in section 4.3. $\mathcal{R}_{\text{succ}}$ is used for the results in figure 3 and table 2. The second reward function we use is defined by

$$\mathcal{R}_{\text{obj}}(E_{\text{opt}}) = \text{relu}(1 - \tau(E_{\text{opt}} - E_0)) \tag{25}$$

The variable $\tau$ is modulated. Starting at $\tau = 0.005$, every 10 epochs it is increased by a factor of 1.5 if $\overline{\mathcal{R}} > 0.5$. The purpose of the relu function is to keep the reward in the range $[0, 1]$ which ensures numerical stability of the optimizer. Additionally, $\tau$ is modulated to try to ensure that the reward signal is strong. More specifically, if $\overline{\mathcal{R}} > 0.5$ then most of the reward values will be clustered at the top of the interval, thus we increase $\tau$ to amplify the signal. Although this reward function doesn't map directly to the relevant performance metric in this case, we use it for the benchmark results in table 1.

# G  DETAILS ON PARAMETER OPTIMIZER (DYNAMIC ANISOTROPIC SMOOTHING)

For parameter optimization we use a zeroth-order evolutionary optimization algorithm based of Reifenstein et al. (2024). This algorithm evolves a distribution of parameters described by two variables $\theta_x \in \mathbb{R}^P$ and $\theta_L \in \mathbb{R}^{P \times P}$. Our goal is to maximize the expected reward function which can be written as

$$\mathcal{R}(\theta_x, \theta_L) = \mathbb{E}_{v, \eta, J}\, \rho(\text{traj}(\theta_x + \theta_L v, \eta, J)) \tag{26}$$

with the expected value take over three distributions. As described in section 3.4, $v$ is a random variable in $N(0,1)^P$ which is mapped to a perturbation in the parameter space by the matrix $\theta_L$. $\eta$ is a random vector corresponding to the stochastic behavior of the trajectory dynamics themselves and lastly $J$ is an instance of the Ising problem chosen from the relevant distribution. DAS works by first computing sample trajectories over these three distributions, and the using the resulting reward values to estimate the gradient of $\mathcal{R}$ with respect to $\theta_x$ and $\theta_L$. The gradients are calculated using the following estimators

$$\frac{\partial \mathcal{R}}{\partial \theta_L} = (\theta_L^{-1})^T \mathbb{E}_{v,\eta,J} \left( vv^\top \rho(\text{traj}(\theta_x + \theta_L v, \eta, J)) - I \right) \tag{27}$$

$$\frac{\partial \mathcal{R}}{\partial \theta_x} = (\theta_L^{-1})^T \mathbb{E}_{v,\eta,J} v \rho(\text{traj}(\theta_x + \theta_L v, \eta, J)). \tag{28}$$

In practice, we estimate this value using a batch of $B$ samples of the random variable $J$, each of which has $R$ independent samples of the random variables $v$ and $\eta$. This results in $BR$ total samples used in the estimation. Or, in other words, at each iteration we use $B$ total problem instances and run $R$ trajectories of the algorithm for each instance to estimate the gradient. This is mainly because parallelization over trajectories of the same instance is a little easier and uses less GPU memory. We typically used $B = 20$ and $R = 400$ for our results.

### G.1 Derivation of Equations equation 28

For the sake of completeness we will include a brief derivation of equation. For explanation of equation equation G.1 (gradient of $\theta_L$) see Reifenstein et al. (2024). To compute the gradient of the smoothed reward function ($\mathcal{R}(\theta_x, \theta_L)$ here and is equivalent to $h(L, x)$ in Reifenstein et al. (2024)) we can first write it as follows

$$\mathcal{R}(\theta_x, \theta_L) = \mathbb{E}_{v,\eta,J}\, \rho(\text{traj}(\theta_x + \theta_L v, \eta, J)) = \int_{v \in R^D} f(\theta_x + \theta_L v) \kappa(v) dv \tag{29}$$

with

$$f(\theta_x + \theta_L v) = \mathbb{E}_{\eta,J}\, \rho(\text{traj}(\theta_x + \theta_L v, \eta, J)) \tag{30}$$

and $\kappa$ being the probability density function of the $\mathcal{N}(0,1)$ variable $v$. Then through the change of variables $u = \theta_x + \theta_L v$, $v = \theta_L^{-1}(u - \theta_0)$ we can write this as

$$\mathcal{R}(\theta_x, \theta_L) = \int_{u \in R^D} f(u) \kappa \left( \theta_L^{-1}(u - \theta_0) \right) \det(\theta_L)^{-1} du \tag{31}$$

then we can take the derivative

$$\frac{\partial}{\partial \theta_x} \mathcal{R}(\theta_x, \theta_L) = \frac{\partial}{\partial \theta_x} \int_{u \in R^D} f(u) \kappa \left( \theta_L^{-1}(u - \theta_x) \right) \det(\theta_L)^{-1} du \tag{32}$$

$$= \int_{u \in R^D} f(u) \frac{\partial}{\partial \theta_x} \kappa \left( \theta_L^{-1}(u - \theta_x) \right) \det(\theta_L)^{-1} du \tag{33}$$

$$= \int_{u \in R^D} f(u)(\theta_L^{-1})^T \kappa' \left( \theta_L^{-1}(u - \theta_x) \right) \det(\theta_L)^{-1} du \tag{34}$$

because $\kappa$ is a gaussian PDF

$$= (\theta_L^{-1})^T \int_{u \in R^D} f(u)\theta_L^{-1}(u - \theta_x) \kappa \left( \theta_L^{-1}(u - \theta_x) \right) \det(\theta_L)^{-1} du \tag{35}$$

$$= (\theta_L^{-1})^T \int_{u \in R^D} v f(\theta_x + \theta_L v) \kappa\,(v)) \, dv = (\theta_L^{-1})^T \mathbb{E}_{v,\eta,J} v \rho(\text{traj}(\theta_x + \theta_L v, \eta, J)) \tag{36}$$

## H    CALCULATION OF TIME TO SOLUTION (TTS) FOR ISING MACHINES

Time to solution (TTS) is defined as the amount of "time" it takes to optimize the given instance with 99% success probability. In this work, we use TTS to compare different Ising machine based algorithms. Because the computation time of all Ising machine algorithms is bottle-necked by the costly matrix vector multiplication that is needed every step of the algorithm, we use TTS in the units of number of steps of the algorithm. This takes out a factor relating to the specific hardware that is used making analysis easier. Thus, TTS is calculated as

$$\text{TTS} = T\frac{\log(1 - 0.99)}{\log(1 - P_s)} \tag{37}$$

where $T$ is the number of steps/iterations, and $P_s$ is the probability of finding the target solution in one run of that many steps.

## I    DETAILS OF TRAINING FOR BENCHMARK RESULTS

### I.1    TABLE 1 RESULTS

For the neural CO benchmark we use an architecture with $T_c = 20$, $D = 3$ and $M = 3$. The number of iterations is set to $T = 300$ for the smaller problem sizes and $T = 1200$ for the larger problem sizes. For the smaller problem sizes ($N = 200$-$300$) we train a network from scratch for 400 epochs with hyper-parameters $R = 400$ and $B = 20$. For the larger problem size ($N = 800$-$1200$) we use the trained parameters for the corresponding smaller problem set and fine tune them on the larger problem set for 200 epochs. We use a training set size of 100 problem instances and a test set size of 1000 problem instances (to be compatible with the results of Sanokowski et al. (2025)). See section J for discussion on why 100 problem instances is sufficient for a training set. We use the objective based reward function for all results on these benchmark (see sec F for details). Because out method has few parameters the training is relatively efficient compared to other methods such as Sanokowski et al. (2025). For example, training the RB-small graphs takes about 4 minutes for our method using an A100 GPU while it is reported that this takes around 16 hours for SDDS in table 7 of Sanokowski et al. (2025).

### I.2    TABLE 2 RESULTS

For the G-set benchmark we use an architecture with $T_c = 20$, $D = 3$ and $M = 3$. The number of iterations is set to $T = N$ in all cases except for the case of the unweighted planar graphs in which it is set to $T = 4N$. The parameters are first tuned on a smaller set of 100 instances of problem size $N = 200$ taken from the same distribution (same graph parameters). Then they are fine-tuned on another set of 100 instances of problem size $N = 800$ generated from the same distribution as the corresponding G-set instances. We use the hyper-parameters $R = 400$ and $B = 20$ and we use the success-rate based reward function for this benchmark (see sec F for details).

## J    IN-DISTRIBUTION GENERALIZATION

In this section we will look at the effect of training set size on test error. In this work we typically use around $\sim 100$ problem instance for training. This may seem like a small number relative to many other machine learning settings, but in our case a small number is sufficient. In figure 7 we show that the test error (shown in solid traces) will be similar to the training error (shown in dashed traces), and overfitting will not happen, as long as there around $\sim 10$ training problem instances. This phenomenon likely depends on the exact distribution of problem instances that we are considering, and reflects the fact that the optimal dynamics required to optimize different instances in the same class are very similar.

## K    DETAILS ON THE EFFECT OF HYPER-PARAMETERS ON PERFORMANCE

In table 3 we show the success rate of cNPIM and dNPIM for different network parameters.

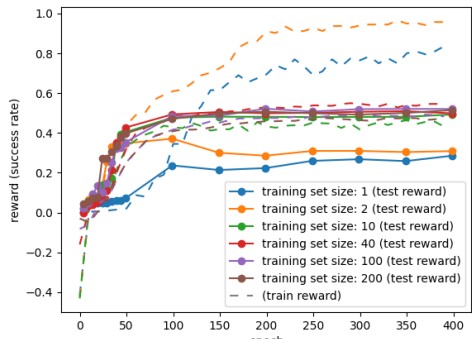

Figure 7: Average train and test reward for different numbers of training problem instances. Problem instances are $N = 100$ SK model.

| $T_c$ | 4 | 8 | 12 | 4 | 4 | 8 | 8 | 12 | 12 | 4 | 4 | 8 | 12 | 8 |
|---|---|---|---|---|---|---|---|---|---|---|---|---|---|---|
| $D$ | 1 | 1 | 1 | 3 | 1 | 1 | 3 | 1 | 3 | 3 | 3 | 3 | 3 | 3 |
| $M$ | 1 | 1 | 1 | 1 | 3 | 3 | 1 | 3 | 1 | 3 | 5 | 3 | 3 | 5 |
| **total P** | 6 | 10 | 14 | 16 | 16 | 28 | 28 | 40 | 40 | 46 | 76 | 82 | 118 | 136 |
| **cNPIM** | 0.183 | 0.350 | 0.351 | 0.179 | 0.283 | 0.514 | 0.445 | 0.518 | 0.425 | 0.390 | 0.549 | 0.538 | 0.542 | **0.550** |
| **dNPIM** | 0.032 | 0.221 | 0.237 | 0.070 | 0.094 | 0.272 | 0.277 | 0.309 | 0.276 | 0.270 | 0.282 | 0.303 | **0.310** | 0.306 |

Table 3: Table showing the effect of network architecture on performance for $N = 100$ SK problem instances. Equivalent data is shown in figure 3c as well. The average success rate for $N = 100$ SK problem instances is shown for different architectures parameterized by the three network hyperparameters.

## L   G-SET TTS DETAILS

In table 4 we show the time to solution of different algorithms with respect to each individual G-set instance. The units of time-to-solution are number of iterations to solution since the compute intensive matrix vector product is the computational bottleneck for each algorithm. The cut values we use are the same as that of Goto et al. (2021) and are, to the best of our knowledge, the best cut values for these instance that have been published for these instances. Our method finds the same cut value in all cases we have checked. We also include an additional result for instance G22 of size $N = 2000$ to show that these results likely scale to larger instances as well.

| | Graph Type | NPIM TTS | SOTA TTS | NPIM/SOTA | Cut Value Found | Best Known Cut Value |
|---|---|---|---|---|---|---|
| G1 | N=800, R, + | **2.55e+04** | 6.01e+04 | 0.42 | 11624 | 11624 |
| G2 | N=800, R, + | **2.28e+05** | 9.20e+05 | 0.25 | 11620 | 11620 |
| G3 | N=800, R, + | **5.63e+04** | 1.70e+05 | 0.33 | 11622 | 11622 |
| G4 | N=800, R, + | **1.00e+05** | 2.09e+05 | 0.48 | 11646 | 11646 |
| G5 | N=800, R, + | **2.11e+05** | 2.26e+05 | 0.93 | 11631 | 11631 |
| G6 | N=800, R, +/- | **3.52e+04** | 1.04e+05 | 0.34 | 2178 | 2178 |
| G7 | N=800, R, +/- | **4.84e+04** | 1.46e+05 | 0.33 | 2006 | 2006 |
| G8 | N=800, R, +/- | **6.55e+04** | 3.59e+05 | 0.18 | 2005 | 2005 |
| G9 | N=800, R, +/- | **1.40e+05** | 2.24e+05 | 0.62 | 2054 | 2054 |
| G10 | N=800, R, +/- | **5.51e+05** | 6.22e+05 | 0.88 | 2000 | 2000 |
| G11 | N=800, T, +/- | **2.86e+04** | 2.22e+05 | 0.13 | 564 | 564 |
| G12 | N=800, T, +/- | **5.51e+04** | 7.86e+04 | 0.70 | 556 | 556 |
| G13 | N=800, T, +/- | **2.74e+05** | 3.73e+05 | 0.74 | 582 | 582 |
| G14 | N=800, P, + | 1.66e+08 | **1.31e+07** | 12.67 | 3064 | 3064 |
| G15 | N=800, P, + | 9.75e+06 | **4.63e+05** | 21.07 | 3050 | 3050 |
| G16 | N=800, P, + | 3.32e+07 | **4.94e+05** | 67.07 | 3052 | 3052 |
| G17 | N=800, P, + | 5.53e+07 | **3.09e+06** | 17.89 | 3047 | 3047 |
| G18 | N=800, P, +/- | **5.01e+05** | 5.08e+05 | 0.99 | 992 | 992 |
| G19 | N=800, P, +/- | **1.48e+05** | 1.80e+05 | 0.82 | 906 | 906 |
| G20 | N=800, P, +/- | **1.17e+04** | 4.24e+04 | 0.28 | 941 | 941 |
| G21 | N=800, P, +/- | **2.61e+05** | 5.74e+05 | 0.45 | 931 | 931 |
| G22 | N=2000, R, + | **9.77e+05** | 2.56e+06 | 0.38 | 13359 | 13359 |

Table 4: Instance-wise TTS of different methods on G-set graphs.

