# OpenReview forum: "Neural Network Ising Machines: Algorithm Unrolling for Combinatorial Optimization"
_ICLR.cc/2026/Conference — Submitted to ICLR 2026_

### Official Review · Reviewer_GHVz · 2025-10-17

**Soundness:** 2
**Presentation:** 3
**Contribution:** 3
**Rating:** 6
**Confidence:** 3

**Summary:**

This paper introduces a novel neural network-based method inspired by Ising machines for solving combinatorial optimization problems. The approach is trained using a zero-order optimization method proposed by Reifenstein et al. (2024). The authors benchmark their method against recent state-of-the-art (SOTA) techniques from both the unsupervised combinatorial optimization and Ising machine communities, demonstrating strong performance across a wide range of problems. Additionally, the paper includes ablation studies examining problem difficulty, parameter count, and problem size.

**Strengths:**

- **Novelty:** The proposed method appears to be original and distinct from existing approaches.
- **Performance:** The method achieves competitive results on well-established benchmarks.
- **Rigor:** The inclusion of ablation studies provides valuable insights into the method’s robustness and sensitivity to key variables.

**Weaknesses:**

- **Lack of Clarity on Optimization:** The paper employs a zero-order optimizer from Reifenstein et al. (2024), but the algorithm itself is not explained. A brief introduction or intuition about the optimizer would significantly improve accessibility for readers unfamiliar with the reference.
- **Reward Function Assumptions:** The reward functions in Appendix F relies on knowledge of the optimal energy value. This assumption raises concerns about fairness in comparisons with methods that do not use such information.

**Questions:**

1. **Gradient Derivation (Q1):**
   Could the authors provide an intuitive explanation of the principles underlying the gradient derivations in Appendix G? This would enhance the paper’s clarity and make it more self-contained.

2. **Reward Function Justification (Q2):**
   How is the use of the optimal energy value in the reward justified? Since other methods do not rely on this information, it is unclear whether the comparison is equitable. A discussion on this point—and an exploration of the method’s performance using only the raw energy as a reward—would be highly informative.

3. **Architectural Alternatives (Q3):**
   The MLP parametrization resembles recurrent architectures like LSTMs. Have the authors explored using LSTMs or similar architectures instead? If so, how does their performance compare to the proposed MLP-based approach?

4. **Clarification on Table 1 (Q4):**
   What does “top 30” refer to in Table 1? Could the authors please clarify?

5. **Diversity of the solutions (Q5):**
If I understand correctly in each trajectory the best solution is taken and for example in Table 1 then the average of these solutions is computed. Is there some diversity in the best solution between trajectories or do all trajectories propose give the same best solution?

---

> ### Author Response · Authors · 2025-11-20
>
> **Response:**
>
> Thanks for your comments and questions and we appreciate the reviewer for taking the time to read the manuscript in detail.
>
>
> **Lack of Clarity on Optimization:**
>
> Thanks for the comment, we will add clarity on the optimization process and equations.
>
>
> **Reward Function Assumptions:**
>
> This is a really good observation, we have added text to clarify this more. You are right that we wouldn’t want to assume prior knowledge about the optimal objective of the instances. What we do is as follows: a training instance set is generated in which we assume that we don’t know the optimal objectives of. However, during the training process as we run many trajectories we keep track of the best objective value found for each instance. Over the course of training once the algorithm begins to be effective this target objective will stabilize to some value which we believe in most cases it the optimal objective value (or at least it matches the solution found by other algorithms). So, no previous knowledge about the problem solutions is needed during training.
>
>
> **Gradient Derivation (Q1):**
>
> The equations for the gradient derivation come from two things. First, in zeroth-order optimization one typically uses a gradient estimator which comes from taking small random perturbations to the parameter space. If we take this randomly perturbed parameter and weight it by the resulting reward function then averaging these over many trials will give us an approximation of the gradient. The reason for the $\theta_L^{-1}$ in equations 27 and 28 is because the perturbations we are taking are scaled by a matrix $\theta_L$, so we need to “divide by” this perturbation strength to get the true gradient. This is related to the classic finite difference equation for the gradient $ \frac{f(x+h) - f(x)}{h}$ except instead of 1/h we have a matrix $\theta_L^{-1}$ to take into account the estimated local curvature of the objective function. We will add additional details on these equations and how they are derived.
>
>
> **Reward Function Justification (Q2):**
>
> Yes, as we mentioned earlier this optimal energy value is not assumed to be a pre-known optimal energy, but is taken to be the best energy found so far for the particular instance during training. Since every instance is seen many times during training, during the later stage of training this objective value will be close to the optimal (or at least the best energy that can be found by the tuned algorithm). We will make sure to clarify this in the paper and the appendix.
>
>
> **Architectural Alternatives (Q3):**
>
> You are right that there are some similarities between LSTMs and our MLP parameterization. We did briefly consider more advanced sequential modelling architectures for our method such as LSTMs (or maybe even transformers), however we limited our study to only very simple architectures for the following reasons. First of all, because we are using a zeroth-order optimization method the number of parameters of our model is somewhat limited (this was touched upon in the conclusion as a potential drawback of our approach). So, more complicated architectures such as LSTMs would be difficult to train. Secondly, we also want the model to be relatively lightweight so that it doesn’t incur an additional cost. That said, I think you are right that LSTMs are simple enough models (and similar enough to what we are already doing) that it would be possible in principle to use our approach with this sort of architecture. This could be interesting because the LSTM could potentially learn something like “resetting” that is sometimes used in CO where the algorithm would remember a previous good solution and then reset back to it when the system is stuck. Although we cannot answer this at the moment, this would be an interesting thing to explore in a future work so we appreciate your comment on this.
>
> **Clarification on Table 1 (Q4):**
>
> “Top 30” refers to the fact that we run 30 trajectories and take the best solution found out of them. Because our model is more lightweight this takes the same computational cost of other models (such as SDDS) for a single run so this is why we use it for comparison. The time reported is the time it takes to run all 30 trajectories for all instances. We will make sure to clarify this in the text, thanks for bringing this up.
>
> **Diversity of the solutions (Q5):**
>
> This is a good question, and the answer is yes, there is diversity. This relates to our answer to Q4 in which we use the top solution out of 30 trajectories (if all solutions were the same this would be pointless). This can also be seen visually in the lower left panel of figure 2 in which the energies of multiple trajectories are shown. For both sets of parameters (encoded in blue and red) we can see that different energies are achieved for many trajectories.

---

> > ### Comment · Reviewer_GHVz · 2025-11-22
> > **Answer to Rebuttal**
> >
> > I appreciate the authors’ thorough and detailed response. To further enhance the clarity and impact of the paper, I encourage them to incorporate the discussed details into the manuscript. Given that my concerns have been satisfactorily addressed, I have revised my score to 8.

---

### Official Review · Reviewer_6UyN · 2025-10-30

**Soundness:** 4
**Presentation:** 3
**Contribution:** 3
**Rating:** 4
**Confidence:** 3

**Summary:**

This paper proposes a learning-based approach for combinatorial optimization via learning the parameters of an iterative dynamical system (Ising machines).  The major contributions of this work are in the combination of algorithmic unrolling, ising machines, and zeroth-order optimization in the context of CO problems.  Computationally, once trained, their approach achieves strong results on benchmark instances for max independent set, max clique, and max cut CO problems.

**Strengths:**

- **Novelty**:  This paper proposes a novel approach that combines algorithmic unrolling, ising machines, and zeroth-order optimization in the context of CO problems.  While this is a combination of existing frameworks, I believe this combination is sufficiently novel and quite refreshing from recent approaches that primarily utilize RL/transformer-style rollouts.
-  **Results**: On the instances evaluated, this approach performs quite well, with the ability to compute high-quality solutions relatively quickly.

**Weaknesses:**

- **Adaptability**: From my understanding, this approach is relatively limited in terms of the classes of optimization problems that it can be used on, e.g., those that can be formulated as Equation (1).  Compared to other exact/heuristic/learning-based methods, this is a relatively strong limitation in terms of applicability.
- **G-Set Results**: The authors compute time-to-solution (TTS) on the G-set benchmarks using reference cut values drawn from prior Ising-machine literature rather than from the globally best-known Max-Cut results reported in combinatorial-optimization studies. While this choice maintains consistency with neural Ising comparisons, it also means that the reported TTS values correspond to approximate rather than truly optimal targets. For this reason, it would be more informative to include information on the differences in solution quality and time compared to the best-known approaches.
- **Scalability**: The experiments are limited to G-set instances up to $N=1000$, with no results reported for the larger, more challenging graphs. The authors state that the scalability is with respect to model size, rather than CO problem size, so I am not sure why they would limit their evaluation to small instances.  The absence of results on larger instances makes it difficult to assess how well the method scales relative to state-of-the-art Ising and Max-Cut solvers.  This is a further concern given the limited applicability to other classes of problems. Additionally, there is no reporting of training time, which makes the scalability of training unclear.

**Questions:**

**Questions**
- [1] propose an approach based on learning and Ising machines for CO.  Can the authors detail the differences in these works and include this in the paper?
- How long do these methods take to train, especially compared to other methods, e.g., DiffUCO?  These should all be included in the appendix.
- How does the performance of a model generalize out-of-distribution?
- Can the authors provide more information on the Gurobi results, i.e., optimality gaps and the time Gurobi takes to find equivalent quality solutions (when Gurobi finds better solutions)?  Furthermore, was Gurobi run with MIPFocus=1 (to prioritize primal solutions)?  If not, this should be done, given the heuristic focus of this work.

**Remarks**:
- In the abstract and throughout the paper, the authors constantly state that they are "solving" instances.  This needs to be changed since their method is a heuristic, and solving should be reserved for exact methods.
- Figure 7 "training training" should be "training".

**References**:
- [1] Bo Lu, Yong-Pan Gao, Kai Wen, and Chuan Wang. Combinatorial optimization solving by coherent
ising machines based on spiking neural networks. Quantum, 7:1151, 2023.

---

> ### Author Response · Authors · 2025-11-20
>
> Thanks for your comments and questions and we appreciate the reviewer for taking the time to read the manuscript in detail.
>
> **Adaptability:**
>
> Although this point is not emphasized in the main text, the appendix discusses how the framework can be generalized beyond the specific formulation we study. We focus on Ising/Max-Cut because it is well established and serves as a starting point for developing and evaluating new combinatorial optimization algorithms. This practice is standard in the field, as reflected in prior works such as [1][2].
>
> As described in appx. D, we believe our method is very adaptable to many types of optimization problems because the $h_i$ that is input to the MLP is essentially a discrete gradient. Problems like SAT and MIP can thus be easily applied in this framework.
>
> [1] Zhang et al. (2023). Let the Flows Tell
>
> [2] Sanokowski et al (2025). Scalable Discrete Diffusion Samplers: CO and Statistical Physics.
>
> **G-set Results:**
>
> As we stated in the text, we use the best known solution found from previous results (which we also find). The best known solutions for these instances are often found by Ising machine algorithms and previous authors have also run traditional heuristics on the same instances (for example see [1]). So, we are technically comparing against many other algorithms including non-Ising-machine algorithms like breakout local search [2].
>
> Regarding the inclusion of solution time and quality on G-set instances, it is possible to include this as well at TTS but we are not sure if it is a very useful metric in this case. The same optimal solution is found by our method for the instances considered. It is thus more informative to simply include the probability of finding this good solution (which is encoded in TTS). For the sake of clarity, we have added in Table 4 the reference best known cuts and cuts found by our approach and shown that they match for all N=800 instances.
>
> [1] Goto et al (2021). High-performance CO based on classical mechanics.
> [2] Benlic et al (2013). Breakout Local Search for the Max-Cut problem.
>
> **Scalability:**
>
> You are right that scalability is a problem, as scalability is a problem with any CO algorithm. What we mean by this is that we believe that the scalability of our technique is on-par with existing Ising machine algorithms that are not explicitly data-driven, as shown in figure 3a.
>
> The results contained in Table 4 of the appendix show that the best known solution is found for all G-set instances of size N=800, which adds more evidence that our approach scales well compared to other methods that only find approximate solutions for the same instances, such as the paper you have referenced [Lu et al 2023].
>
> To confirm this further, we tested our method on the G22 instance of size N=2000. This instance was also used in the paper you referenced [Lu et al 2023] however the authors’ method did not find the best known cut value of 13359 reported in [1]. Our method, once trained, finds this optimal cut value of 13359 with a TTS of ~1.7 seconds on an A100 GPU instance. So we believe that our method is likely scalable on these instances as well. We added this result to Table 4 in the appendix.
>
> And again, regarding the comment on limited applicability we believe that our algorithm is easily generalizable to other problems as we touched upon in our earlier comment.
>
> [1] Goto et al (2021).
>
> **Q1:**
>
> Thanks for showing us this work. We will add a reference to it in the final version of our manuscript, if accepted. My understanding is that the authors propose a few modifications of the original CIM equations. They modify the dynamics (shown in eq 1)  and propose an adaptive annealing schedule of one of the parameters (shown in fig 4e). They find that making these adjustments to the existing equations can improve performance of the algorithm. These sorts of incremental improvements to IM dynamics are very common in the literature.
>
> The idea behind this submission is that instead of iteratively making adjustments by hand to equations and annealing schedules, we can have the dynamics be fully parameterized by a NN and then tune this NN from scratch (using zeroth-order optimization). We find that this approach outperforms even the SOTA hand-crafted IMs such as SBM and CAC. We believe this is a significant result worth publishing and is relevant for both IM and neural CO communities.
>
> **Q2:**
>
> This is a good question, we added an additional sentence to section I.1 of the appendix which touches on this. The training time is significantly less than other algorithms.
>
> **Q3:**
>
> This is addressed in section 4.4 and figures 3a and 3d.
>
> **Q4:**
>
> The results for Gurobi were obtained from the tables 1 and 2 of [1]. The parameter MIPfocus=1 is set, see details of the Gurobi setup at https://github.com/ml-jku/DIffUCO/blob/main/DatasetCreator/Gurobi/GurobiSolver.py.
>
> [1] Sanokowski et al (2025).
>
> **Remarks:**
>
> Thanks for these comments we will make these changes.

---

### Official Review · Reviewer_SQVt · 2025-10-31

**Soundness:** 1
**Presentation:** 1
**Contribution:** 1
**Rating:** 2
**Confidence:** 3

**Summary:**

This paper studies Ising machines as optimizers for combinatorial problems—specifically Max-Cut, Maximum Independent Set (MIS), and Max-Clique—and proposes a zeroth-order (gradient-free) optimization approach for tuning/steering the machine. The authors compare against prior Ising-based methods and aim to demonstrate improved solution quality and/or efficiency.

**Strengths:**

The Ising formulation is a natural modeling choice for quadratic objectives, and the paper targets three canonical NP-hard problems with broad interest.

Using zeroth-order optimization is well-motivated in settings with noisy or non-differentiable hardware, and the paper’s perspective could be useful to practitioners working with analog or black-box solvers.

The manuscript attempts to position the work within the growing literature on physical/Ising-style optimizers for CO, which is timely.

**Weaknesses:**

Evaluation metrics (TTS vs objective quality). The paper emphasizes “time to solution” (TTS), defined as the time required to reach a solution with 99% success probability. While TTS is common in annealing/Ising communities, it is less standard in the combinatorial optimization literature, which typically leads with objective quality (cut value, clique size, independent set size), approximation ratios or normalized optimality gaps, and then reports wall-clock time. I encourage the authors to complement TTS with conventional CO metrics. This would make results easier to compare with non-Ising baselines.

**Questions:**

None

---

> ### Author Response · Authors · 2025-11-20
>
> **Response:**
>
> Table 1 is solely focused on objective quality and wall-clock time, presented exactly as in prior CO work, which directly contradicts the reviewer’s statement that conventional CO metrics are absent.
>
> The opening of Section 5 explicitly motivates the importance of reporting both TTS and objective-quality metrics.
>
> Given that the reviewer’s sole criticism is contradicted by the content of the paper, we invite the AC to verify this by examining Table 1 and Section 5, and to assess the weight that this review should carry in evaluating the submission.

---

> > ### Comment · Reviewer_GHVz · 2025-11-22
> > **Comment to Review of Reviewer SQVt**
> >
> > I would like to clarify that the authors do, in fact, report the objective solution quality of their algorithm—as well as those of other algorithms—in Table 1 of the manuscript. This directly addresses the reviewer’s concern, which appears to be based on an oversight.

---

### Comment · Area_Chair_f8Fg · 2025-11-24
**Discussion Period**

Dear reviewers,

The discussion period is now open. Please use the “Official Comments” to engage in discussions about each other's reviews and the authors' rebuttal, and update your assessments or comments as appropriate.

Did the authors' rebuttal adequately address your concerns? We kindly ask that you update your reviews based on these discussions and your evaluation of the rebuttal, even if your overall assessment remains unchanged.

Thank you all for your contributions.

Best regards, AC

---

### Meta-Review · Area_Chair_ycus · 2026-01-07

**Summary:**

This paper proposes a learning-based Ising machine for combinatorial optimization via algorithm unrolling and zeroth-order optimization. The approach is technically sound and demonstrates competitive results on several benchmarks. However, reviewers raised concerns regarding applicability and scalability, including evaluation on larger instances and the clarity of training cost claims. In addition, a procedural issue regarding a page-limit violation in the initial submission was raised by both reviewers and the Area Chair.

**Reviewer Concerns:**

Reviewers raised concerns regarding applicability and scalability, as well as a procedural issue related to page-limit compliance. While the authors addressed several technical clarification requests in the rebuttal, including evaluation metrics, optimization details, and additional large-instance results, concerns about broader applicability beyond Ising/Max-Cut formulations and scalability to more diverse problem instances remain only partially resolved. The page-limit violation in the initial submission was clarified but remains a relevant procedural consideration.

**Reviewer Scores:**

One reviewer revised their score upward after the rebuttal, indicating that their technical concerns were adequately addressed. Other reviewers would likely remain borderline or negative due to persistent concerns about applicability and scalability. Overall, the discussion did not result in a clear positive consensus.

---

### Decision · Program_Chairs · 2026-01-26

Reject